# TEM image restoration from fast image streams

Håkan Wieslander[1]*, Carolina Wählby[1,2], Ida-Maria Sintorn[1,3]

**1** Department of Information Technology, Uppsala University, Uppsala, Sweden, **2** BioImage Informatics Facility of SciLifeLab, Uppsala, Sweden, **3** Vironova AB, Stockholm, Sweden

* hakan.wieslander@it.uu.se

**Data Availability Statement:** All images files are available from https://zenodo.org/record/4113244.

**Funding:** Financially supported by the Swedish Foundation for Strategic Research www.strategiska.se (grant BD150008), the European Research Council www.erc.europa.eu (grant

## Abstract

Microscopy imaging experiments generate vast amounts of data, and there is a high demand for smart acquisition and analysis methods. This is especially true for transmission electron microscopy (TEM) where terabytes of data are produced if imaging a full sample at high resolution, and analysis can take several hours. One way to tackle this issue is to collect a continuous stream of low resolution images whilst moving the sample under the microscope, and thereafter use this data to find the parts of the sample deemed most valuable for high-resolution imaging. However, such image streams are degraded by both motion blur and noise. Building on deep learning based approaches developed for deblurring videos of natural scenes we explore the opportunities and limitations of deblurring and denoising images captured from a fast image stream collected by a TEM microscope. We start from existing neural network architectures and make adjustments of convolution blocks and loss functions to better fit TEM data. We present deblurring results on two real datasets of images of kidney tissue and a calibration grid. Both datasets consist of low quality images from a fast image stream captured by moving the sample under the microscope, and the corresponding high quality images of the same region, captured after stopping the movement at each position to let all motion settle. We also explore the generalizability and overfitting on real and synthetically generated data. The quality of the restored images, evaluated both quantitatively and visually, show that using deep learning for image restoration of TEM live image streams has great potential but also comes with some limitations.

## Introduction

In recent years there has been an explosion in the amount of image data generated from microscopy experiments due to faster and more automated microscopes. The increased speed at which data is collected results in data-volumes that are costly to handle and easily exceed the computational and data storage resources available. This entails a need for smarter acquisition and analysis methods. This is especially apparent for Transmission Electron Microscopy (TEM) where the difference between sample size and details of interest is much larger than in other types of microscopy. As a comparison: in histopathology whole slide imaging (WSI) a sample is typically $2 \times 1cm$ and imaged with a pixel size of $0.275\mu m$ resulting in approx. 4 Giga

ERC2015CoG 682810), (grants awarded to CW) and the Uppsala University AI4Research initiative (awarded to IMS). The funders had no role in study design, data collection and analysis, decision to publish, or preparation of the manuscript. Vironova provided part-time salary and instrument access (TEM) for author IMS. The funder provided support in the form of salaries for authors IMS but did not have any additional role in the study design, data collection and analysis, decision to publish, or preparation of the manuscript. The specific roles of these authors are articulated in the 'author contributions' section.

**Competing interests:** The authors have declared that no competing interests exist. The commercial affiliation of author IMS does not alter our adherence to PLOS ONE policies on sharing data and materials.

pixels. Whereas, in TEM a sample is typically $3mm$ in diameter and the details of interest often require at least a $1nm$ pixel size resulting in more than 4 Tera pixels. Hence, one TEM sample contains more than 1000 times more data than one WSI sample if the full sample is imaged at full resolution. It is thus impossible to image the whole sample at the resolution required for detailed analysis when working with TEM. Since not all data contains valuable information data acquisition and analysis should focus on the data that has the best possibility of answering the research question at hand. One approach to tackle this issue is to use a hierarchical image acquisition and analysis approach, i.e. to acquire and analyze low resolution images to detect regions of interest (ROI) for high resolution imaging. Automatic detection of ROIs can then be used in an online feedback fashion to reduce the amount of data collected. Online feedback microscopy has shown great potential in fluorescence microscopy to automatically pinpoint locations of interest during acquisition [1]. Conrad et al. [2] presented a workflow for high-resolution image acquisition based on a low resolution pre-scan for automatic identification of mitotic cells. For TEM, Kylberg et. al. [3] and Suveer et al. [4] proposed schemes for hierarchical imaging and analysis workflows for virus and cilia detection in TEM, respectively.

A typical approach when working with TEM is that a pathologist/microscopist uses the motorized stage to move the sample at low magnification and manually/visually scans for ROIs. These regions are then imaged at a higher magnification. This approach is time and expert demanding. In addition, capturing a high quality image requires pausing the movement momentarily (typically 1-2 sec) to ensure all movement has stopped prior to acquiring the image. Due to these pauses it can take several hours to automatically image a sample at low magnification where sufficient detail is visible to detect promising objects or ROIs for high magnification imaging. On the other hand, capturing the images while scanning over the sample is fast but results in images with a lot of motion blur and noise, making them unsuitable for further analysis. However, if this noise and blur could be removed, faster imaging and analysis would be possible.

In the field of image restoration, image deblurring refers to the task of recovering the underlying sharp image representation from a degraded image. The degradation process can be formulated as

$$y = h * x + \eta \tag{1}$$

where $y$ is the captured blurry image, $h$ is the blur kernel, $x$ is the underlying sharp image and $\eta$ is the noise term. Hence, blur can be seen as a combination of degradation based on both intrinsic factors, such as camera focus, and external factors e.g. motion and noise, that mainly arise from the image acquisition process. The recovery process is thus the inverse problem where either $h$ is known a priori and $x$ is recovered, or both $x$ and $h$ are recovered blindly. The effect of the noise term must also be minimized in the process, and depending on the type of noise, different methods can be applied [5]. This results in an ill-posed problem as the number of unknown factors exceeds the data space. Solutions usually involve adding constraints or priors to reduce the solution space. The deblurring problem has been studied extensively and proposed solutions date back to the 1970s, the most prominent of which are based on either Bayesian inference with Maximum Likelihood Estimations [6, 7] or deconvolution in Fourier space [8]. These methods, however, usually produce overly smooth images or images with ringing artifacts. Methods have been proposed to tackle these challenges, for instance, Yuan et al. suggested an extension to the Richardson-Lucy algorithm utilizing a multi-scale approach with bilateral filters [9].

A challenge with deconvolution solutions is that they are sensitive to the approximation of the point spread function (PSF). These functions easily become complex, especially for TEM

data where moving the microscope at a high magnification results in irregular vibrations and drift. Added on top of this is a composition of noise factors. Another problem for deconvolution based solutions is that they usually involve an iterative scheme which makes them rather slow. They also assume that the noise level is low and often struggle with saturated pixels [10]. Recently, more data driven approaches have been developed, based on Convolutional Neural Networks (CNNs), which have shown great promise in dealing with complex PSFs. A benefit of CNNs is that the deblurring procedure is performed directly on the input intensities, mapping the input to the output through a complex (neural network) function, hence approximating blur kernels and iterative deconvolutions can be avoided. Recent research has also suggested using a combination of CNNs with more classical approaches. For instance, Xu et. al [10] propose to construct the CNN in such a way that it can learn and perform deconvolution based on weighted sums of separable 1D filters. Their method is however limited by the need to train a separate network for each blur kernel. Ren et al. [11] overcome this limitation by calculating a generalized low-rank approximation to a large number of blur kernels which is then used to initialize the parameters of the network. These methods deal with the non-blind deblurring case, where the blur kernels are known or constructed a priori. For blind deblurring, Chakrabarti [12] suggest a CNN to predict the complex discrete Fourier transform coefficients for a deconvolution filter. The deconvolution can then be performed in the frequency domain. Similarly, Sun et al. [13] propose a method for removing non-uniform motion blur based on predicting the blur kernel for smaller image patches resulting in a field of non-uniform blur kernels which can later be used to deconvolve the image. Several other success stories for blind deblurring involve Generative Adversarial Networks (GANs). One example is Kupyn et al. [14] who propose an end-to-end learned conditional GAN combined with perceptual loss [15] resulting in state of the art performance. This is extended in [16] where the backbone is replaced by a flexible feature pyramid network to attain a balance between performance and efficiency.

A parallel field to single image deblurring is video and burst image deblurring. When multiple frames are available, methods can be designed to make use of the additional data in neighboring frames to guide the restoration. This can be achieved either with or without alignment of the frames. Without alignment the network has the freedom to extract relevant spatial/temporal information from each of the input frames, for instance through a series of down-convolutional layers as in [17], through 3D-convolutions as in [18], or as in [19] through attention modules for efficient handling of different amounts of blur. Other methods use alignment modules within the network to guide the restoration. Zhao and Zhang et al. [20] propose Filter Adaptive Convolutional Layers to align features from the previous time step with the current time step. Wang et al. [21] perform the alignment of feature maps through deformable convolutions and achieved state of the art results in the NTIRE19 [22] challenge. These types of networks have some drawbacks when the translation between frames is large and are hence not studied here since the translation between frames in the evaluated datasets is irregular and sometimes too large for valuable information to be present in a small time window. If the input images are (more or less) aligned, the network aligning steps can be avoided and focus can be put on the actual deblurring task. Aitalla et al. [23] propose a network, operating on a burst of images, based on copies of the same UNet architecture with weight sharing and exchange of information through global max pooling layers (where features are pooled with the max operation between networks).

Motion deblurring has mainly been studied with natural scene images and often by simulating the blur occurring through human movement of a camera. Within microscopy, motion deblurring is not such a well-explored area. However, image restoration with focus on denoising (not motion deblurring) has shown great promise for electron microscopy, using both

classical (non deep learning) methods [24] and unsupervised as well as semi-supervised deep learning based methods [25]. Another closely related problem, that of sliding the sample past the camera to create an overview image through video mosaicing, has been more extensively studied. For instance [26] explore this for confocal microscopy for *in vivo* imaging, but solve the issue with blurry frames by excluding them. Alternatively, [27] study real time video mosaicing for high-resolution microendoscopy and solve the blurring issue with a low camera exposure time and slow translations over the sample.

In this paper we study the opportunities and limitations of deep learning methods applied to image reconstruction from fast (live stream) TEM image acquisition. The purpose is to understand if and how a live image stream can be used to replace the standard slow acquisition of low magnification images currently used to find ROIs for high magnification imaging. To the best of our knowledge, no prior work in video deblurring exists for TEM imaging. Video deblurring for TEM poses a number of challenges. Firstly, the magnification at which the specimen is imaged is highly sensitive to movements and vibrations from the mechanics and the piezo electric elements in the stage. Secondly, like with all types of microscopes, noise will always be present in the images, the amount of which is determined by the exposure time and illumination. Lastly, the electron induced changes in sensitive samples can pose challenges for image registration. Here, we focus on the image restoration side of this problem, and investigate how deep learning can be used to improve the quality of images corrupted by motion blur. We also explore the generalizability of networks to non-seen textures and the networks' prediction and overfitting characteristics.

Our approach to do this is to compare and contrast the performance of two CNN architectures for single-image deblurring and burst image deblurring, respectively, on two transmission electron microscopy live image stream datasets. We adapt the best performing architecture—the burst architecture—to the challenges inherent to TEM data to further improve the performance. Along with real data, we also use synthetic data to visualize and explore the model's and training scheme's effect on the restoration result. The paper is structured as follows: First, the image capturing process and dataset design is described in the Data section. Then, the two basic network architectures along with the evaluation metrics used are described in the Method section. In the Experiment section, the training procedure, model comparisons and model improvements are described, along with how we explore and visualize model behaviour, overfitting and generalizability. Finally, we summarize our findings in the Conclusion section.

The main contributions of this paper are: A proof of principle of using CNNs for image restoration from fast TEM video streams. A structured comparison and nuanced evaluation of two video deblurring CNN approaches on real TEM image streams. An application adaptation of the loss function and further improvement of the superior burst architecture. An approach for using synthetic data for visualization of architectural/parameter/training effects. And finally, we design and make available two novel TEM datasets of real video streams and matching high quality images.

## Data

The data was collected using a MiniTEM microscope (Vironova AB). The samples were placed on a regular TEM grid (a regularly spaced metal grid) before imaging (Fig 1). High quality frames of size 2048 x 2048 were first collected with an overlap of 50%. After this, a video sequence of the same area but with a frame size of 1024 x 1024 pixels (the motion going vertically upwards), was collected. All images were captured at a field of view of $32\mu$m, and with a per image exposure time of 15ms. Both high quality frames and blurry frames where saved

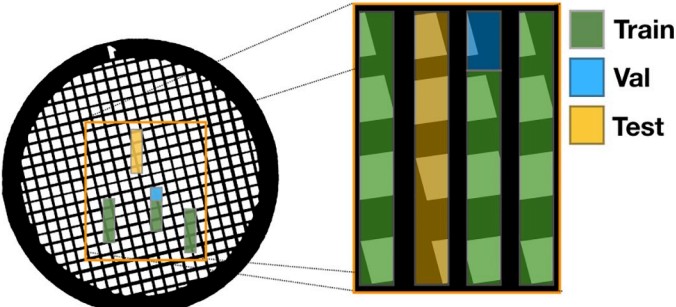

**Fig 1. Illustration of the image capturing process and how the data was divided into training, validation and test sets.** A sample was imaged by moving the sample past the microscope in a straight vertical movement at four different locations. Note that the placements in the figure does not exactly correspond with the exact locations where the data was collected.

with 16 bit resolution. Two samples where imaged, one of a calibration grid and one of a kidney sample (Fig 2). The data is made available online at [28]. A total of four imaging sequences were captured per sample. For the kidney sample each imaging sequence contained 7-23 high quality images and for the calibration grid 14-18 high quality images. One of the imaging sequences (for each dataset) was put aside and used for testing. As a validation set, one grid cell was chosen that had no overlap with the rest of that image sequence. For the kidney sample the test set resulted in 6 high quality images and the validation set in 3 high quality images. For the calibration grid 8 high quality images were used for testing and 2 for validation. The respective training set size for the kidney dataset and calibration grid dataset was 38 and 35 high quality images. Fig 1 shows how the data was divided into training, validation and test sets.

The frames from the video sequence and high quality images were matched by first manually finding which blurry frame best matched each high quality image. The blurry frames were then registered to the high quality image in a semi-automatic way via enhanced correlation coefficient maximization [29], where initial translation was manually set. The number of frames for both datasets was set to 5, i.e., for each high quality image, 5 blurry frames were

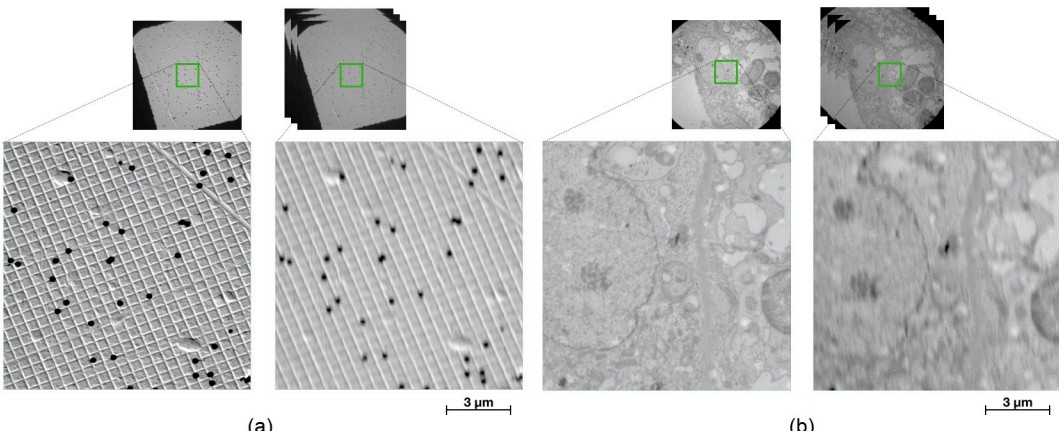

**Fig 2.** Example images from the two datasets with high quality images (left image of each pair) together with the corresponding blurry and noisy images of the same region (right image of each pair) (a) calibration grid, (b) kidney.

registered. This was chosen so that the overlap between frames would not be too small, since after registration we extract the intersection between overlapping regions.

To reduce the range of intensities, the data was scaled to 8 bits. The maximum and minimum values for each dataset were manually set based on the histograms and structures in the sample. The maximum was chosen based on the brightest regions in the samples and set slightly above these intensities. The minimum intensity was chosen so that the black regions between grid cells become zero but dark regions within the sample still maintained some intensity variations.

For each image, a mask was created to indicate the actual sample (content in grid cells). This was used to crop patches containing only information of the sample, leaving out the background resulting from imaging the metal grid between the grid cells.

The mask was created by applying a threshold at zero (minimum value after normalization). For the training data, the mask was thereafter eroded by an circular structuring element of radius 128 pixels for 2 iterations to make it possible to make random crops of size $128 \times 128$ px containing no grid pixels. Since there is an overlap in the high quality data, the bottom half of each image used for testing and validation was discarded. This was done so that multiple images of the same FOV would not be present. The test and validation images were then regularly tiled (128x128), and tiles including the grid were discarded.

## Synthetic data for visualizing network performance

It can be difficult to visualize the actual effect a network has on a sample if it is very complex. We thus created three synthetic test images. One image simulates some of the structures we see in the kidney sample by bright and dark spots and lines in different directions. A structured repetitive pattern (mimicking the squares in the calibration grid) is simulated by drawing some lines at an angle on a blank image. We create two such images, one with only lines in one direction and one with lines in two directions. This to see how dependent the network actually is on the texture of the sample. We add synthetic motion to the images and Gaussian noise to match the real data (the amount was determined visually), see Fig 3.

## Methods

### Model selection

To find a baseline for model/method selection we evaluated two different network architectures. The first one, DeblurGANv2 [16], (from here on referred to as DeblurGAN) is based on single image deblurring and uses three registered frames (RGB) as input. It uses a pre-trained backbone to extract features at multiple scales which are then fused and up-scaled to a single output. The network is trained with an adversarial loss function based on both a "patch" discriminator, evaluating smaller patches of the image, and a "full" discriminator evaluating the full image.

The second architecture, proposed by Aitalla et al. [23], (from here on referred to as Burst) is based on the premise that the slight variation in blur and noise between frames can be utilized to make a better restoration. Collecting images in a "burst" can thus help restoring the underlying information. This network consists of copies of the same UNet architecture with tied weights. The copies exchange information through "global max pooling" layers where features are pooled with the max operation between the networks.

Non-learning methods based on PSF approximation are not considered here. The main reason for this exclusion being that the approximation of the PSF becomes highly difficult with the irregular motion characteristics present in the dataset.

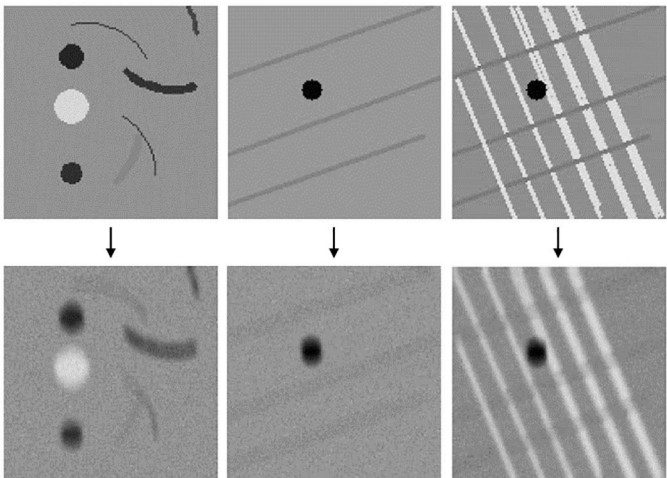

**Fig 3. Synthetic data for evaluating network performance.** The top row shows one image with bent lines and circular spots of different intensities and two images with lines in different directions. The bottom row shows the same images after convolution with a motion blur kernel followed by the addition of Gaussian noise.

## Evaluation metrics

The networks were evaluated based on the Structural Similarity Index Measure (SSIM) [30] and Peak Signal-to-Noise Ratio (PSNR). SSIM is a windowed approach evaluating the structural similarity between two sub-images (windows) $x$ and $y$. The similarity is measured as:

$$SSIM(x, y) = \frac{(2\mu_x\mu_y + c_1)(2\sigma_{xy} + c_2)}{(\mu_x^2 + \mu_y^2 + c_1)(\sigma_x^2 + \sigma_y^2 + c_2)}, \tag{2}$$

where $\mu_x$ and $\mu_y$ are the averages of the respective windows, $\sigma_x, \sigma_y$ the variances and $\sigma_{xy}$ the covariance. The terms $c_1$ and $c_2$ are used to stabilize the division and are derived from the dynamic range of the respective images.

PSNR measures the ration of the maximum power of a signal in relation to the corruption error and is calculated as:

$$PSNR = 10 \cdot log_{10}\left(\frac{MAX_I^2}{MSE}\right) \tag{3}$$

where $MAX_I$ is the maximum intensity value the image can take and $MSE$ is the mean squared error between the generated and ground truth image.

Since SSIM and PSNR do not always capture the perceived quality of the restoration, resulting images were also visually evaluated.

## Experiments

### Network training

The networks were trained with training schemes as similar as possible to those presented in their respective papers. If not stated otherwise all parameter settings were the same as those proposed in each paper. For DeblurGAN, the network was first trained with an ImageNet pre-trained backbone frozen for 3 epochs and then unfrozen until convergence. The loss function was a mixture of the adversarial RaGAN-LS loss, mean squared error as content loss and a

**Table 1. Quantitative performance on the two different datasets for the two different network architecture.**

| Network<br>Dataset | No deblurring | | Burst | | DeblurGAN | |
|---|---|---|---|---|---|---|
| | SSIM | PSNR | SSIM | PSNR | SSIM | PSNR |
| Kidney | 0.77 ± 0.051 | 29.29 ± 2.71 | 0.85 ± 0.045 | 31.98 ± 3.61 | 0.82 ± 0.055 | 31.78 ± 2.30 |
| Calibration grid | 0.56 ± 0.071 | 21.58 ± 0.94 | 0.89 ± 0.036 | 28.23 ± 1.69 | 0.84 ± 0.042 | 26.46 ± 1.46 |

perceptual loss based on VGG19 feature maps. For the Burst network the loss was a combination of L1-loss and un-normalized horizontal and vertical gradients.

Both networks were trained for 2000 epochs and the network parameters of the best epoch (determined on a validation set and measured in SSIM) were saved for evaluation on the test set. DeblurGAN was trained with Adam as an optimizer with a learning rate of $10^{-3}$. The Burst network used the same optimizer but with an initial learning rate of 0.003 and a per epoch learning rate decay of 0.99 (slightly higher than proposed in the original paper to attain a more stable training curve).

For DeblurGAN, three adjacent registered frames were used as input, scaled to an intensity range of 0—1, and the output of the network was a single channel gray scale image. For the Burst network, the number of input frames was varied between 3 and 5. The network was evaluated with 5 frames as input.

## Network architecture comparison

The two network architectures were compared quantitatively, by evaluating their performance and qualitatively, by visual inspection of the outputs on both datasets where the networks are trained on the same data as tested on. The quantitative results (mean and standard deviation) are presented in Table 1. A qualitative comparison of the results from the respective networks is presented in Figs 4 and 5 (best viewed on a digital screen).

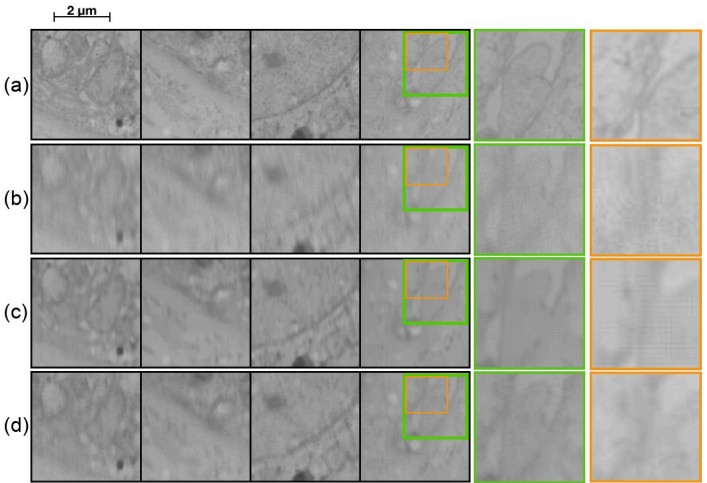

**Fig 4. Qualitative evaluation of the different networks on the kidney dataset.** (a) Ground truth, (b) Input, (c) Output from Burst network, (d) Output from DeblurGAN. The images in the rightmost column are displayed with increased brightness for better visualization.

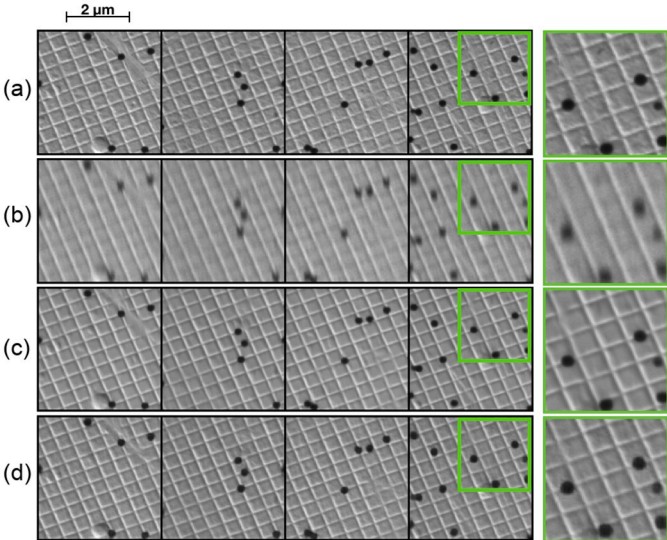

**Fig 5. Qualitative evaluation of the different networks on the calibration grid dataset.** (a) Ground truth, (b) Input, (c) Output from Burst network, (d) Output from DeblurGAN.

From Table 1 we see that the Burst network performs slightly better than the DeblurGAN network. However, visually (Fig 4) we can see that both the DeblurGAN and the Burst network improve upon the input but with varying levels of quality and with differing side effects. The Burst network outputs a smoother image but with a grid-like artefact pattern. This pattern is an effect of the transpose convolutions in the network. The DeblurGAN, on the other hand, creates a fake texture on top of the output image. This is most probably an effect of the adversarial training where by "hallucinating" certain textures the network can better fool the discriminator. None of the networks are capable of resolving the finer details visible in the ground truth. See for instance the leftmost column of Fig 4, where in the ground truth there are several small fine lines that are not present in the output. The problem probably lies in the fact that there is no information in the input data pertaining to these fine structures. Since the networks are limited to working with the data given as input, one can achieve a better but limited representation of the input, or the network has to compensate and make up information. This could explain the artefacts seen in the DeblurGAN output, where textures that should not be there are created—the generator overcompensates to better fool the discriminator.

The results look much more promising on the calibration grid data (Fig 5) where both networks manage to reconstruct the "horizontal" lines that are barely visible in the input data. We can also see this in the quantitative results in Table 1 where there is a larger improvement on the calibration grid data as compared to no deblurring, than for the kidney data.

## Model improvements

As the Burst architecture performed slightly better than the DeblurGAN architecture we chose to continue running experiments on this network to explore and suggest improvements. We started with replacing the UNet base with a Dense UNet [31]. The Dense UNet has shown great potential in other image restoration tasks [32], and we therefore decided to incorporate it in the Burst model and evaluate its performance. Based on its dense connections, it iteratively concatenates feature maps, and can hence be more parameter efficient since all layers can access the preceding layers' feature maps. The dense blocks consist of several convolutional

blocks including batch-normalization—ReLU—convolution- dropout operations that are densely connected. See Fig 6 for an illustration of the proposed model architecture. The dense UNet model was built with 4 dense blocks in contracting and expanding path where each block consists of 4 convolutional blocks.

To get rid of the grid-like pattern seen in Fig 4, we replaced the transpose convolutions with a nearest neighbour upsampling followed by a convolution [33] (referred to as 'Transition up' in Fig 6). The alternative up-sampling method Pixel-shuffle [34] was investigated but did not further improve the results (not presented).

The combination of the transitional layers and the gradient based loss function produces another artifact (textural "wobbly" grid pattern), seen best in Fig 7c. We therefore replaced the gradient loss with an SSIM based loss function, which has shown to be beneficial for image restoration tasks [35].

Experiments including global max pooling layers also in the decoder (results not presented) showed no significant improvements on the reconstructed image (neither visually nor quantitatively), and hence these additional layers were not included.

We trained the new network with Adam as optimizer with a fixed learning rate of 0.001 for 2000 epochs. We fixed the seed to have the same data sampling and network initialization for all experiments.

Table 2 shows the quantitative performance for the model including the proposed improvements (individually trained on each dataset). Note that the performance increase with the SSIM loss could be due to using the same evaluation metric as optimization score. A qualitative comparison, showing the benefit of the improvements can be seen in Fig 7. The abbreviations for network configuration stands for (BU) Burst with UNet base, (BDU+T) Burst with Dense

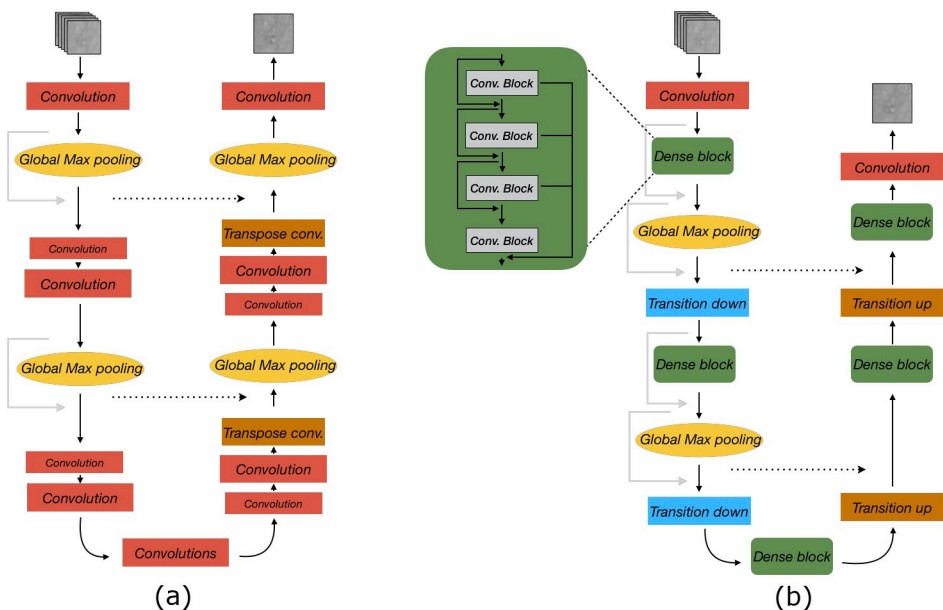

**Fig 6. Illustration of Burst model and the proposed improvements.** (a) Original Burst (UNet base) network. Note that image is only for illustration of differences and that the model also employs other layers such as batch norm and activations. See [23] for full model configuration. (b) The proposed Burst (Dense UNet base) network. In comparison to the UNet model, the convolutional layers in the proposed model have been replaced by dense blocks and global max pooling layers are removed in the decoder path. Instead, there are 'Transition up' layers consisting of a nearest neighbour upsampling followed by a convolution and 'Transition down' is a spatial max pooling operation. Note that the Global Max pooling is the layer that combines data from the 'burts' of multiple input images.

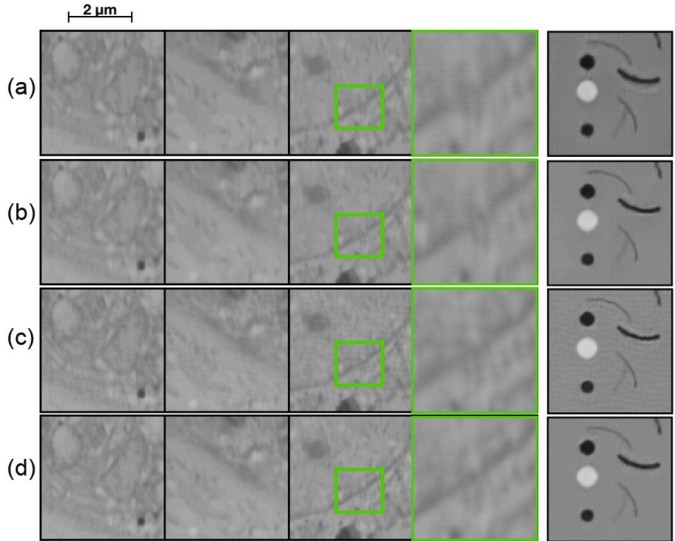

**Fig 7. Qualitative comparison of network improvements.** From top (a) BU, (b) BDU + T, (c) BDU + UC, (d) BDU + UC + SSIM. Green square (zoomed in area) highlights an area where improvements on fine details can be seen.

UNet base using transpose convolutions, (BDU+UC) Burst with Dense UNet replacing transpose convolution with upsampling followed by convolution and (BDU+UC+SSIM) Burst with Dense UNet base, upsampling followed by convolutions and SSIM loss.

The effect of the improvements are shown in Fig 7. By replacing the UNet base with a Dense UNet we can see a slight improvement in the amount of motion the network removes (see zoomed in area of in Fig 7). The effect of replacing the transposed convolutions and changing the loss function is best seen in the synthetic image. Both Fig 7a and 7b have a grid-like pattern in the output image. For Fig 7a and 7b this arises from the transposed convolutions. Fig 7c has a different more textural "wobbly" pattern which is a result of the gradient loss with the transition layers. In Fig 7d it is clear that both of these effects have been removed.

## Exploring network behaviour and overfitting

To better understand the behaviour of the model we evaluate the networks trained on the two different datasets on the synthetically generated images. To adjust the network to the slightly different intensity distribution we do approximately 20 forward passes to update the running mean and variance in the batch normalization layers (no backward pass, i.e., no updating of the weights). The results are shown in Fig 8. Note that the networks are trained on real data and only evaluated on synthetic data.

**Table 2. Quantitative evaluation of network improvements.**

| Dataset<br>Configuration | Kidney | | Calibration Grid | |
|---|---|---|---|---|
| | SSIM | PSNR | SSIM | PSRN |
| No deblurring | 0.77 ± 0.051 | 29.29 ± 2.71 | 0.56 ± 0.071 | 21.58 ± 0.94 |
| BU | 0.85 ± 0.045 | 31.98 ± 3.61 | 0.89 ± 0.036 | 28.23 ± 1.69 |
| BDU+T | 0.86 ± 0.045 | 32.65 ± 2.66 | 0.90 ± 0.027 | 29.22 ± 1.40 |
| BDU+UC | 0.85 ± 0.047 | 33.22 ± 2.52 | 0.90 ± 0.022 | 28.92 ± 1.38 |
| BDU+UC+SSIM | 0.87 ± 0.048 | 33.17 ± 2.02 | 0.90 ± 0.025 | 29.10 ± 1.42 |

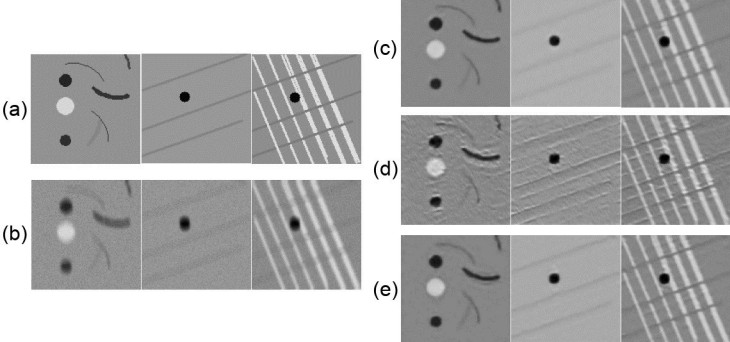

**Fig 8. Resulting output on synthetic images when the network is trained on different types of data.** From top (a) Ground truth, (b) Input, (c) Trained on kidney data, (d) Trained on calibration grid data, (e) Trained on both types of data.

From all three output images the majority of the motion appears to have been removed, i.e the network has learnt some sense of motion. However, the network trained on the calibration grid data has also learnt the underlying grid-like pattern. In the two rightmost images in Fig 8d, we see the effect where the network adds "horizontal" lines to the image that are not present in the input. This result indicates that the network has over-fitted to the underlying texture. The texture/patterns in the kidney data is much more complex, making it more difficult for the network to over-fit to this specific texture, and provides some explanation for the lower quantitative performance on this dataset (see Table 2).

### Exploring network generalizability

Since the two datasets were captured under similar imaging conditions, and should have approximately the same amount of motion blur and noise, we further tested the network's generalization behaviour by training on one dataset and evaluating on the other. Quantitative results for between sample generalizability are presented in Table 3 and qualitative results in Fig 9. From the same experiment, results on the synthetic images are presented in Fig 8.

We also evaluate the model to model generalization across different splits of the data. For each split one of the imaging sessions were held out for testing. Quantitative cross validation results are presented in Table 4.

The results in Table 3 show that there is a drop in performance when testing on a different sample type than trained on. The largest drop occurs when testing on calibration grid data. The performance drops with an average of 0.13 points between the network trained on kidney data and the network trained on calibration grid data. The effect can be seen in Fig 8 where the network trained on the calibration grid images learns the underlying texture. This over-fitting to the underlying texture can also be seen in Fig 9d (see arrows) where one can see ghost

**Table 3. Quantitative comparison when switching training data and test data.**

| Test data / Training data | Kidney | | Calibration grid | |
|---|---|---|---|---|
| | SSIM | PSNR | SSIM | PSNR |
| Kidney | 0.87 ± 0.048 | 33.17 ± 2.02 | 0.77 ± 0.039 | 25.04 ± 0.98 |
| Calibration grid | 0.83 ± 0.042 | 31.39 ± 3.22 | 0.90 ± 0.025 | 29.10 ± 1.42 |
| Both | 0.85 ± 0.044 | 31.48 ± 3.42 | 0.88 ± 0.028 | 28.04 ± 1.26 |

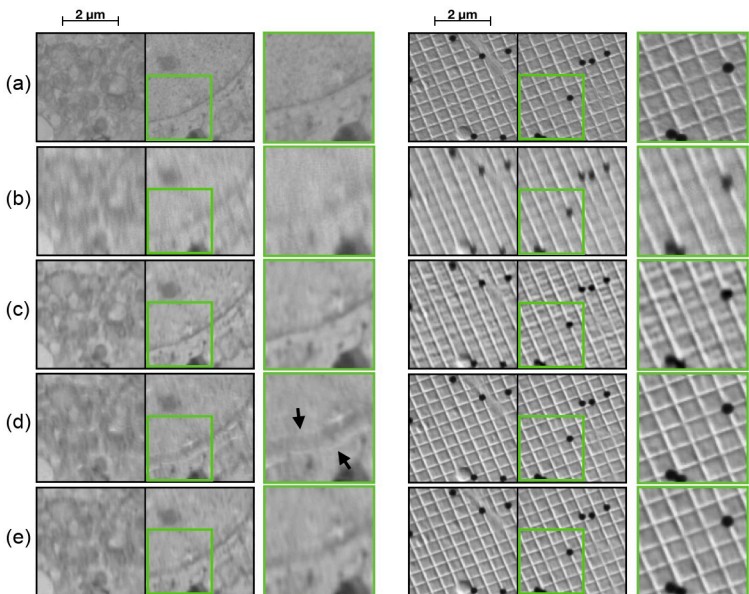

**Fig 9. Resulting output when the network is trained on different types of data.** From top (a) Ground truth, (b) Input, (c) Trained on kidney data, (d) Trained on calibration grid data, (e) Trained on both types of data.

**Table 4. Network performance across different splits of the data.**

| | Cross validation Splits | | | | Kidney | | Calibration Grid | |
|---|---|---|---|---|---|---|---|---|
| | **1** | **2** | **3** | **4** | **SSIM** | **PSNR** | **SSIM** | **PSNR** |
| 1 | Test | | | | 0.87 ± 0.04 | 31.26 ± 3.86 | 0.88 ± 0.03 | 28.16 ± 1.05 |
| 2 | | Test | | | 0.83 ± 0.04 | 32.53 ± 1.69 | 0.90 ± 0.03 | 29.10 ± 1.42 |
| 3 | | | Test | | 0.87 ± 0.05 | 33.17 ± 2.02 | 0.88 ± 0.06 | 28.16 ± 2.27 |
| 4 | | | | Test | 0.86 ± 0.06 | 33.68 ± 2.71 | 0.88 ± 0.04 | 28.26 ±1.51 |
| | | | | Average: | 0.86 ± 0.05 | 32.66 ± 2.70 | 0.89 ± 0.04 | 28.42 ± 1.62 |

artifacts, which are also apparent in Fig 8d on synthetic images. However both networks remove some motion and noise even on an unseen sample type. This demonstrates that the network learns a combination of the motion blur/noise and the underlying texture. When training a network on both datasets we see a slight decrease (on average) in performance as compared to having one network per sample (Table 3). Fig 8e also shows that the over-fitting to texture is less prominent and the network generalizes slightly better between the two samples.

The results in Table 4 show that the network generalizes well between different splits of the data and hence network performance is not restricted to one specific split of the data.

## Conclusion

We have shown the potential and limitations of applying deep learning methods to improve the quality of TEM images corrupted by motion blur and noise. This is one step towards efficient automated imaging in TEM. We have studied the quality of the output both quantitatively and qualitatively and conclude that using deep learning for TEM image restoration from

fast image streams shows substantial promise but has some drawbacks that should be considered for each specific application.

The two network architectures explored (both with and without improvements) were able to reduce blur/noise in the input, but with varying degrees of success. However, the image quality improvements are limited to the information present in the input data. If the information, for instance about finer structures, is lost due to the degradation process in the image acquisition stage, it is difficult for the network to reconstruct it despite having multiple views of the scene. Either the networks try to compensate for this and make up information that is not present in the input, or become limited to reconstructing fine details in the input. We could for instance see this effect in Fig 4 where the DeblurGAN network makes up a significant amount of texture that is not there (and should not be there), while the Burst network outputs a smoother image. The presence of artifacts that the network has created has clear drawbacks when working with pathological diagnostics where the visual details provide clues to the patient's diagnosis and the pathologist must be able to trust the information in the data. The alternative is to obtain an output with more uncertainties in the restorations but no fake details.

The improvements made to the Burst network resulted in improved blur and noise removal. The combination of the dense blocks and the 'global max pooling' layers gives the network the possibility to both access the feature maps from preceding layers (from the dense blocks) and to exchange information between the different feature maps from the multi channel input (through the global max pooling layers). The network becomes more parameter efficient but requires more GPU memory to train.

We have shown that synthetic data is very useful to better visualize and understand the network predictions and performance when working with image restoration tasks. We could for instance discover signs of over-fitting and artifacts arising from the network structure and loss functions. The synthetic data also gave a better visual overview and understanding of what the network had focused on. The network learns a combination of motion and noise and texture/structures in the sample.

We have also shown that one means of reducing the over-fitting to a specific texture is to train with different types of samples.

## Author Contributions

**Conceptualization:** Håkan Wieslander, Ida-Maria Sintorn.

**Data curation:** Ida-Maria Sintorn.

**Formal analysis:** Håkan Wieslander.

**Funding acquisition:** Carolina Wählby.

**Investigation:** Håkan Wieslander, Carolina Wählby, Ida-Maria Sintorn.

**Methodology:** Håkan Wieslander, Carolina Wählby, Ida-Maria Sintorn.

**Project administration:** Carolina Wählby, Ida-Maria Sintorn.

**Resources:** Carolina Wählby, Ida-Maria Sintorn.

**Software:** Håkan Wieslander.

**Supervision:** Carolina Wählby, Ida-Maria Sintorn.

**Writing – original draft:** Håkan Wieslander, Carolina Wählby, Ida-Maria Sintorn.

**Writing – review & editing:** Håkan Wieslander, Carolina Wählby, Ida-Maria Sintorn.

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
