## [Decision Letter · Decision Letter 0]

27 Nov 2020

PONE-D-20-33777

TEM image restoration from fast image streams

PLOS ONE

Dear Dr. Wieslander,

Thank you for submitting your manuscript to PLOS ONE. After careful consideration, we feel that it has merit but does not fully meet PLOS ONE’s publication criteria as it currently stands. Therefore, we invite you to submit a revised version of the manuscript that addresses the points raised during the review process.

We look forward to receiving your revised manuscript.

Kind regards,

Khanh N.Q. Le

Academic Editor

PLOS ONE

Journal Requirements:

'Financially supported by the Swedish Foundation for Strategic Research www.strategiska.se (grant BD150008), the European Research Council www.erc.europa.eu (grant ERC2015CoG 682810), (grants awarded to CW) and the Uppsala University AI4Research initiative (awarded to IS). The funders had no role in study design, data collection and analysis, decision to publish, or preparation of the manuscript.'

We note that one or more of the authors are employed by a commercial company: Vironova AB

Reviewers' comments:

Reviewer's Responses to Questions

**Comments to the Author**

1. Is the manuscript technically sound, and do the data support the conclusions?

Reviewer #1: Partly

Reviewer #2: Yes

2. Has the statistical analysis been performed appropriately and rigorously? 

Reviewer #1: Yes

Reviewer #2: Yes

3. Have the authors made all data underlying the findings in their manuscript fully available?

Reviewer #1: Yes

Reviewer #2: No

4. Is the manuscript presented in an intelligible fashion and written in standard English?

Reviewer #1: Yes

Reviewer #2: Yes

5. Review Comments to the Author

Reviewer #1: • The abstract can be rewritten to be more meaningful. The authors should add more details about their final results in the abstract. Abstract should clarify what is exactly proposed (the technical contribution) and how the proposed approach is validated.

• The paper does not explain clearly its advantages with respect to the literature: it is not clear what is the novelty and contributions of the proposed work: does it propose a new method? Or does the novelty only consist in the application?

• The contributions of the paper are not clearly identified (Section 1, last paragraph). Authors need to be claimed their contributions and justify with sufficient experimental results.

• Bullet your contribution at the end of the introduction section.

• Overall, the Methodology Section needs to extend with more details in each step and I recommend adding an algorithm (pseudocode) for the proposed method.

• Authors need to provide justifications for all the parameters setting.

• More scientific reasoning should be added to the experimental results' explanations.

• Manuscript needs to be thoroughly revised and rewritten in the format of a journal publication and must be edited by a native English speaker.

• Please highlight the advantages and disadvantages of your method.

• I need to see a comparison between the proposed method with other previous methods.

• In results, authors should add the convergence graphs.

• Do the authors employ any cross-validation scheme? Please, provide details about it.

Reviewer #2: This manuscript proposed a fast image stream based transmission electron microscopy image restoration by consideration of motion blur and noise. This work itself is interesting. However, some other problems in the manuscript are still concerned in the attached file.

6. PLOS authors have the option to publish the peer review history of their article (what does this mean?). If published, this will include your full peer review and any attached files.

Reviewer #1: No

Reviewer #2: No

---

## [Author Response · Author response to Decision Letter 0]

13 Jan 2021

Dear Editor and reviewers,

Many thanks for the valuable comments and suggestions for the paper. Although not suggested by the reviewers, the comments made us decide to run one additional experiment to fully explore the technological contributions we have made to adjust video deblurring to fast image streams from TEM. We have made some significant changes to the abstract and final paragraphs of the introduction to better clarify our contributions. Furthermore, we have added a figure to clearly illustrate how we have made further developments to CNNs for motion deblurring. Our responses to the review comments are detailed below.

Reviewer #1:

The abstract can be rewritten to be more meaningful. The authors should add more details about their final results in the abstract. Abstract should clarify what is exactly proposed (the technical contribution) and how the proposed approach is validated.

- Rewrote a major part of the abstract to better highlight proposed method and approach.

The paper does not explain clearly its advantages with respect to the literature: it is not clear what is the novelty and contributions of the proposed work: does it propose a new method? Or does the novelty only consist in the application?

- Made the contribution clearer in the end of the Introduction. The main contribution lies in combining modern techniques for deblurring of TEM data, where the main focus is on the application.

The contributions of the paper are not clearly identified (Section 1, last paragraph). Authors need to be claimed their contributions and justify with sufficient experimental results.

- See previous answer. We have also added the result of one additional experiment to clarify the experimental results.

Bullet your contribution at the end of the introduction section.

- The contributions are now clearly listed in the last paragraph of the introduction.

Overall, the Methodology Section needs to extend with more details in each step and I recommend adding an algorithm (pseudocode) for the proposed method.

- We added a figure of the proposed network configuration along with a more detailed description of our contributions.

Authors need to provide justifications for all the parameters setting.

- We added more details on what parameters are re-used from previous publications and what parameters are optimized for the application presented here. Details regarding training approaches are also clarified.

More scientific reasoning should be added to the experimental results' explanations.

- We have now added an extra experiment and clarified our reasoning regarding model improvements and selection of technical approaches.

Manuscript needs to be thoroughly revised and rewritten in the format of a journal publication and must be edited by a native English speaker.

- The manuscript has been thoroughly revised and edited by a native English speaker.

Please highlight the advantages and disadvantages of your method.

- Added advantages and disadvantages are now better highlighted in the discussion.

I need to see a comparison between the proposed method with other previous methods.

- We now argue about the problems that arise when using more classical methods, like approximating the point spread function for motion deblurring. Also, the problems with using other deep learning techniques that for instance work with time series are discussed, hence motivating why we compare “only” two different network architectures.

In results, authors should add the convergence graphs.

- Since we have so many networks trained, it would take up a lot of space to include all convergence plots and would like to leave these ones out. All networks are trained for a specific number of epochs making sure they have converged.

Do the authors employ any cross-validation scheme? Please, provide details about it.

- We now added cross validation to the manuscript to show that network performance is not restricted to a specific split of the data.

Reviewer #2:

This manuscript proposed a fast image stream based transmission electron microscopy image restoration by consideration of motion blur and noise. This work itself is interesting. However, some other problems in the manuscript are still concerned in the following: 

Grammar mistake in the sentence “shows that using deep learning for image restoration of TEM image live streams has great potential but also carry limitations”.

- We fixed grammar mistakes and let a native English speaker revise the text.

The contribution and innovation of this work should be stated more clearly.

- We made our contributions more clear at the end of the Introduction.

The organization of this manuscript should be added to the end of the introduction.

- We added a description of the organization of the manuscript to the end of the introduction.

More details on the method should be exposed in the text.

- We added an illustration of the model and added text about the content of the components in the model, as well as an additional supporting experiment.

The flow charts of the method should be shown in the manuscript.

- See previous answer.

More evaluation metrics are suggested in the experiments.

- We added PSNR to all experiments.

Could the authors use more kinds of methods in the experiments to validate the methods?

- We added cross validation to validate the results, as well as one more experiment to evaluate the different components of the proposed network improvements.

More works on image restoration should be included, such as “DOI: 10.1016/bs.mcb.2019.05.001”, “DOI:10.1109/TGRS.2014.2307354”, “DOI: 10.1111/jmi.12716”…

- We added 2 of the references. We chose not to include DOI: 10.1109/TGRS.2014.2307354 since it does not relate to motion deblurring or TEM reconstruction.

The conclusion is suggested to be simplified.

- We have re-written the conclusions to make it more clear.

Some references missed the fundamental information, such as volume, number and page. Please check them very carefully.

- We have fixed all errors we could find in the references.

---

## [Decision Letter · Decision Letter 1]

18 Jan 2021

TEM image restoration from fast image streams

PONE-D-20-33777R1

Dear Dr. Wieslander,

We’re pleased to inform you that your manuscript has been judged scientifically suitable for publication and will be formally accepted for publication once it meets all outstanding technical requirements.

Kind regards,

Khanh N.Q. Le

Academic Editor

PLOS ONE

Additional Editor Comments (optional):

Reviewers' comments:

Reviewer's Responses to Questions

**Comments to the Author**

1. If the authors have adequately addressed your comments raised in a previous round of review and you feel that this manuscript is now acceptable for publication, you may indicate that here to bypass the “Comments to the Author” section, enter your conflict of interest statement in the “Confidential to Editor” section, and submit your "Accept" recommendation.

Reviewer #1: All comments have been addressed

Reviewer #2: All comments have been addressed

2. Is the manuscript technically sound, and do the data support the conclusions?

Reviewer #1: Yes

Reviewer #2: Yes

3. Has the statistical analysis been performed appropriately and rigorously? 

Reviewer #1: Yes

Reviewer #2: Yes

4. Have the authors made all data underlying the findings in their manuscript fully available?

Reviewer #1: Yes

Reviewer #2: No

5. Is the manuscript presented in an intelligible fashion and written in standard English?

Reviewer #1: Yes

Reviewer #2: Yes

6. Review Comments to the Author

Reviewer #1: The authors have addressed the reviewer's concerns and the revised version of the manuscript appears to be good.

Reviewer #2: All my questions have been answered. The current version can be published directly in my opinion. Congratulations!

7. PLOS authors have the option to publish the peer review history of their article (what does this mean?). If published, this will include your full peer review and any attached files.

Reviewer #1: No

Reviewer #2: No

---

## [Editor Report · Acceptance letter]

21 Jan 2021

PONE-D-20-33777R1 

TEM image restoration from fast image streams 

Dear Dr. Wieslander:

I'm pleased to inform you that your manuscript has been deemed suitable for publication in PLOS ONE. Congratulations! Your manuscript is now with our production department. 

Kind regards, 

on behalf of

Dr. Khanh N.Q. Le 

Academic Editor

PLOS ONE